# COVID-19 vaccine hesitancy and conspiracy beliefs in Togo: Findings from two cross-sectional surveys

Herve Akinocho[1][☯], Ken Brackstone[2][☯], Nia Eastment[2][☯], Jean-Paul Fantognon[3][☯], Michael G. Head[2,4,5][☯]*

1 Center for Research and Opinion Polls, Lomé, Togo, 2 Clinical Informatics Research Unit, Faculty of Medicine, University of Southampton, Southampton, United Kingdom, 3 Ministry of Health Public Hygiene, and Universal Health Coverage, Lomé, Togo, 4 School of Medicine, University for Development Studies, Tamale, Ghana, 5 School of Public Health, University for Health and Allied Sciences, Hohoe, Ghana

☯ These authors contributed equally to this work.
* M.Head@soton.ac.uk

**Data Availability Statement:** The dataset is openly-available at https://doi.org/10.6084/m9.figshare.23989332.v1.

## Abstract

Togo is a low-income country in West Africa. Estimates from Our World in Data suggest that only 25% of the Togolese population have received at least one dose of any COVID-19 vaccine by June 2023. Whilst the early phase of the pandemic vaccine rollout across 2021 was dominated by higher-income countries taking much of the available supply, there have long been sufficient supplies for all nations. Thus, there remains a need to understand reasons for low uptake in countries such as Togo, here focusing on population confidence and trust, essentially characteristics that could potentially be addressed within health promotion strategies. Two cross-sectional telephone surveys of Togo residents were conducted in December 2020 and January 2022. These surveys asked questions around perceptions of COVID-19, trust in public health messaging, belief in conspiracy theories, and hesitancy around COVID-19 vaccination. Analyses here focus on unvaccinated respondents. Across Survey 1 (N = 1430) and Survey 2 (N = 212), 65% of respondents were men, and 47% lived in Lomé (capital city of Togo). Between Surveys 1 and 2, overall hesitancy (33.0% to 58.0% respectively) and beliefs in conspiracy theories (29% to 65%) significantly increased. Using logistics regression, governmental mistrust was the strongest significant predictor of hesitancy (OR: 2.90). Participants who indicated agreement or uncertainty with at least one conspiracy belief also predicted greater vaccine hesitancy (OR: 1.36). Proactive approaches to public health messaging, that better understand reasons for hesitancy across different demographics, can support uptake of COVID-19 vaccinations within Togo. This includes health promotion campaigns that use locally and nationally trusted knowledge providers (e.g. the health service or religious leaders) for greatest effectiveness at reducing impact of misinformation. Key future research should focus around knowledge gaps and areas of mistrust created by the pandemic, such as the impact of misinformation upon routine immunisation uptake.

**Funding:** This research was funded by the Clinical Informatics Research Unit, at the University of Southampton. Funding was awarded to MGH. The funders had no role in study design, data collection and analysis, decision to publish, or preparation of the manuscript.

**Competing interests:** The authors have declared that no competing interests exist.

## Introduction

The COVID-19 pandemic, caused by a novel coronavirus which spread across multiple countries in early 2020, was likely responsible for an estimated 18.2 millions excess global deaths by the end of 2021 [1]. The sustained spread of the virus had a huge impact upon routine health services globally [2], with extensive and long-lasting socio-economic consequences. Togo is a lower-income country with approximately 8.7m population in West Africa, bordered by Ghana to the west, Benin to the east, Burkina Faso to the north, and the Atlantic Ocean to the south. As of 12 October 2023, the country has reported 39,522 confirmed cases of COVID-19, with 290 confirmed deaths [3]. These numbers are likely to be a significant underestimate, with 7.0 million globally confirmed deaths as of 12 October 2023 [3]. There is evidence showing extensive under-reporting from sub-Sahara Africa (SSA) countries amid a likely higher age-adjusted infection fatality rate [4], and also worsened mortality among hospitalised patients [5]. However, it would still appear that most countries on the African continent did not experience high overall death rates observed in high-income settings such as the United Kingdom or United States of America. There are likely to be many contributory factors to the lower mortality in West Africa, including a younger population, improved preparedness [6], and better outbreak management [7].

By mid-2023, there is still a significant global burden of COVID-19, but it is more manageable. Most countries are looking toward a "post-pandemic" environment, with case management of COVID-19 integrated within routine health systems. However, population-level COVID-19 vaccine uptake remains low across many lower-income countries, with estimates that only 25% of the Togolese population have received at least one dose of any COVID-19 vaccine (by June 2023) [3]. Whilst the early phase of the pandemic vaccine rollout across 2021 was dominated by higher-income countries taking much of the available supply [8], there have long been sufficient supplies for all nations. Thus, there remains a need to understand reasons for low uptake in countries such as Togo. This can help us to learn lesson from the COVID-19 pandemic, here developing a deeper understanding of hesitancy and mistrust ahead of any future similar public health emergency.

Vaccine hesitancy is defined by the World Health Organization (WHO) as the delay in the acceptance, or blunt refusal of, vaccines. Vaccine hesitancy in West Africa has been associated with governmental dissatisfaction and mistrust, particularly of government messaging, for example a boycott of the polio vaccine in Northern Nigeria in 2003–2004 [9]. More recent surveys administered in sub-Saharan Africa (SSA) countries such as Malawi, Mali, and Nigeria, found that dissatisfaction with the government´s response to the COVID-19 pandemic predicted hesitancy [10]. This pattern was also found in Ghana, where supporters of opposition political parties demonstrated greater hesitancy to receive the COVID-19 vaccine than supporters of the party currently in government [11]. Afrobarometer surveys found that Togolese citizens distrust in their government to ensure the safety of vaccines has decreased during the pandemic (62% in January 2021 to 47% in March 2022), but were overall satisfied with the government's handling of the COVID-19 crisis [12,13]. Thus, individual differences in political trust and mistrust may be associated with vaccine hesitancy in Togo. Studies have also shown that COVID-19-related misinformation is common in West Africa. For example, a nationwide survey conducted in Ghana found that over 50% of citizens believed, or expressed uncertainty, that COVID-19 was a biological weapon designed by the Chinese government, and that the virus was designed specifically to reduce or control the population [11]. Other widely-circulating conspiracy theories included (incorrectly) associating 5G telenetwork technology in the spread of COVID-19, and that the pandemic was a plague caused by sins and disbelief in human beings.

Previously, our team published two vaccine hesitancy studies in Ghana, one of which took place electronically across four time points [11], and a one-off survey carried out in-person in a rural location [14]. To date, there has been little COVID-19 research focusing on Togo. Additionally, there is little research more generally focusing around localised beliefs of conspiracy theories and levels of trust, and how this impacts upon vaccine confidence. This study covers descriptions of vaccine hesitancy from two national surveys with Togo citizens. A portion of the Survey 1 data has been made available to policymakers previously for their consideration and decision-making [15]. Here, we provide further insight into both surveys, seeking to identify characteristics associated with vaccine hesitancy using socioeconomic and demographic variables, and to describe knowledge and attitudes toward, and the impact of, the COVID-19 pandemic in Togo. To widen access to our results within Togo and other French-speaking nations, the manuscript is also available in French (S1 Text, translated by author HA).

## Methods

### Design, participants, and procedure

Two nationally representative telephone surveys were administered (S2 Text). Survey 1 was conducted across 1–16 December 2020 –approximately 6 months after the first case of COVID-19 was reported in Togo and prior to any vaccines being available in-country. Survey 2 was implemented across 11–28 January 2022, approximately 9 months after COVID-19 vaccines began to be rolled out in Togo. Thus, the viewpoints described in this research cover the important timepoints of pre- and post- pandemic vaccine rollout. Inclusion criteria included those resident in Togo, aged 18 years or over, who had capacity and were capable of providing informed consent. Exclusion criteria included those who were not resident in Togo, unable to provide informed consent, or were aged under 18 years.

Prior to data collection, a power analysis was conducted to determine the appropriate sample size. We assumed a confidence level of 95%, and a margin of error of approximately 3–5%, and found that the necessary sample was between 385 and 1067 participants in each survey. We achieved these sample sizes in both surveys. However, when considering specifically the participants who were unvaccinated in Survey 2, we fell short of the required number.

Togo cell phone numbers have 8 digits. The first two numbers are specific to each of the telecoms providers within the country. The Togo team generated 6 other random numbers to make up a potential phone number, and these numbers were dialled to make an introduction. Numbers of calls were weighted according to the market share of each cell phone network.

### Ethical approval

Participants provided verbal informed consent prior to taking part in the survey. They were provided with verbal participant information during the initial telephone call and offered written information via email or private message. They were then given time to consider whether they wished to take part. All participants confirmed that they were over the age of 18 before proceeding with the survey. See S1 Fig for a map of Togo, showing the regions. The study received ethical approvals from the Togo Bioethics Research Committee (reference 006/2020/CRBS). Data storage was electronic, on password-protected devices. Personal identifiers such as participant name or address was not collected.

### Measures

**Vaccine hesitancy.** In Survey 1, participants were asked: "When the COVID-19 vaccine becomes available to you, would you like to get vaccinated?" (yes, no, I don´t know). In Survey

2, participants initially indicated whether they had previously received any doses of the COVID-19 vaccine. Among participants who indicated that they had not received any doses, participants were subsequently asked: "When the COVID-19 vaccine becomes available to you, would you like to get vaccinated?" (Yes, No, I don´t know).

**Governmental mistrust.** Participants next indicated the extent in which they agreed with the statement: "I have trust in the Togolese government's response to the COVID-19 pandemic" (1 = *strongly disagree*; 5 = *strongly agree*; *M* = 3.90; *SD* = 1.11). Mistrust was coded by dichotomising participants´ responses (strongly disagree, somewhat disagree, or undecided).

**Conspiracy beliefs.** Participants indicated whether they believed in eight COVID-19-related conspiracy beliefs recorded to be circulating in SSA [11]. They selected "yes" if they agreed with the belief, "unsure" if they were uncertain about the belief, or "no" if they did not agree with the belief (e.g. "To the best of your knowledge. . . [COVID-19] is designed to reduce or control the population".

**Sources of information.** Participants were presented with several places where they may have sought out COVID-19-related information and the pandemic response. These included traditional news sources (TV/radio), the Ministry of Health and health service, government officials, and the internet (e.g. news websites, blogs, Google). Participants selected the sources that they typically used to receive information about COVID-19 and vaccines.

**Demographic variables.** Finally, participants indicated their age (coded into <40 and 40>], gender, religion (Christianity, Muslim, other, none) and marital status (never married, cohabitation without marriage, married, separated but not divorced, divorced, widow/widower). Socioeconomic variables covered education (high [university degree or higher] and low [senior secondary or lower]) and region (Lomé, Maritime, Plateaux, Central, Kara, Savannah) (supplementary).

## Data management and analysis

Data were examined for errors, cleaned, and exported into IBM SPSS Statistics 28 for further analysis. Descriptive statistics summarized respondents' socio-demographics. Inferential statistics were conducted in three phases. First, temporal trends in hesitancy and population prevalence were compared between each survey. Due to the World Health Organization's definition of vaccine hesitancy as the delay in the acceptance, or the blunt refusal of, vaccines [X], hesitancy was coded by dichotomising participants' responses (no, I don't know) to the question: "When the COVID-19 vaccine becomes available to you, would you like to get vaccinated?" Chi-Square $\chi2$ tests were conducted to assess for categorical differences in hesitancy rates and conspiracy beliefs between Surveys 1–2. Descriptive analyses were also conducted to summarize conspiracy beliefs and self-reported sources of vaccine-related information. Next, variable selection was performed using a series of bivariate regressions. The multivariate model consisted of all factors with P values equal to, or smaller than, 0.2 in the bivariate analysis. All assumptions–including tests for confounders, multicollinearity, and outliers–were tested for during model-building. Thus, any analyses that we report passed assumption checks. A combined logistic regression considered all the predictors in a single model, providing the strictest test of potential associations with vaccine hesitancy. Vaccine hesitancy and its associated predictors were rescaled to 0 or 1, which allowed for direct comparison of effect sizes. Participants who provided missing responses were excluded in the subsequent analyses to maintain data quality.

**Table 1. Descriptive statistics of participants from 2020 and 2022 surveys.**

| | Combined (N = 1642) | 2020 (N = 1430) | 2022 (N = 212) |
|---|---|---|---|
| | % (n) | | |
| **Gender** | | | |
| Men | 66.0 (1084) | 66.0 (944) | 66.0 (140) |
| Women | 34.0 (558) | 34.0 (486) | 34.0 (72) |
| **Age** | | | |
| < 40 | 66.2 (1087) | 64.5 (922) | 77.8 (165) |
| 40 > | 33.8 (555) | 35.5 (508) | 22.2 (47) |
| **Marital status** | | | |
| Single/separated | 33.2 (544) | 31.1 (444) | 47.2 (100) |
| Married/in a relationship | 66.8 (1098) | 68.9 (986) | 52.8 (112) |
| **Region** | | | |
| Central | 7.9 (129) | 7.6 (109) | 9.4 (20) |
| Kara | 8.3 (136) | 8.8 (126) | 4.7 (10) |
| Lomé | 47.4 (778) | 49.2 (704) | 34.9 (74) |
| Maritime | 18.0 (295) | 16.1 (230) | 30.7 (65) |
| Plateaux | 12.7 (209) | 12.3 (176) | 15.6 (33) |
| Savannah | 5.8 (95) | 5.9 (85) | 4.7 (10) |
| **Highest education** | | | |
| Senior secondary or lower | 75.5 (1237) | 75.9 (1085) | 71.7 (152) |
| Higher | 24.7 (405) | 24.1 (345) | 28.3 (60) |
| **Religion** | | | |
| Christian | 69.2 (1135) | 70.0 (999) | 64.2 (136) |
| Muslim | 16.3 (267) | 16.2 (231) | 17.0 (36) |
| Other or none | 14.5 (238) | 13.8 (200) | 18.9 (40) |
| **Care responsibilities** | | | |
| No | 27.2 (446) | 24.4 (348) | 46.2 (98) |
| Yes | 72.8 (1194) | 75.6 (1082) | 53.8 (114) |
| **Get vaccinated?** | | | |
| No | 26.6 (437) | 25.7 (368) | 32.5 (69) |
| I don't know | 8.9 (146) | 6.4 (92) | 25.5 (54) |
| Yes | 64.5 (1059) | 67.8 (970) | 42.0 (89) |
| **Sources of information** | | | |
| Mass media (TV/radio) | 94.4 (1550) | 97.4 (1393) | 74.1 (157) |
| THS or health workers | 58.2 (955) | 63.6 (909) | 21.7 (46) |
| Government officials | 33.4 (549) | 37.0 (529) | 9.4 (20) |
| Internet | 25.6 (421) | 26.7 (382) | 18.4 (39) |

## Results

### Participants and socio-demographic characteristics

Table 1 presents descriptive statistics of unvaccinated participants from Survey 1 ($N$ = 1430) and Survey 2 ($N$ = 212). The majority of participants across both surveys were men (66.0%) vs. women (34.0%; $Mage$ = 35.74; $SD$ = 12.25; $Range$ = 18–84). The majority of participants across surveys lived in Lomé (47.4%) and Maritime (18.0%). Further, 24.7% completed higher education compared to 75.5% who completed to senior secondary or lower, and 66.8% reported being married or in a relationship compared to 33.2% who were not in a relationship. By

religion, 69.2% of participants were Christian compared to Muslim or other/none (16.3% and 14.5%, respectively). Finally, 72.8% reported care responsibilities for under 18s or older adults.

## Sources of COVID-19 vaccine-related information

The most commonly accessed sources of COVID-19 vaccine-related information were mass media (e.g. newspapers, radio, TV; 94.4%), the health service (58.2%), government officials (33.4%), and the internet (e.g. Google, news websites, blogs; 25.6%).

## Conspiracy beliefs

Overall, 50.9% (554/1088) of participants indicated agreement with at least one conspiracy belief ($M = 0.57$, $SD = 1.03$; Table 2). The most commonly believed conspiracy beliefs included: ". . . is plague caused by sins and disbelief of human beings" (261/1642; 15.9%), ". . . a biological weapon designed by the government of China" (157/1642; 9.6%), and ". . . designed to reduce or control the population" (151/1642; 9.2%). A Pearson´s Chi-Squared test revealed a significant association between time and conspiracy beliefs, in which the proportion of respondents who indicated agreement increased from 29.2% (CI: 26.6%-31.8%) in Survey 1 (December 2020) to 64.6% (95% CI: 57.9%-71.3%) in Survey 2 (January 2022; $\chi2 (1) = 103.85$, $p < .001$; Fig 1).

Next, 48.4% (794/1642) of participants indicated uncertainty about at least one COVID-19-related conspiracy beliefs ($M = 2.42$, $SD = 2.99$). The most common conspiracy beliefs included: ". . . a virus designed by the pharmaceutical industry to sell their drugs" (601/1642; 36.6%), ". . . is a biological weapon caused by the US government" (572/1642; 34.9%), and ". . . designed to reduce or control the population" (555/1642; 33.8%). A Pearson´s Chi-Squared test revealed a significant association between time and conspiracy beliefs, in which the proportion of respondents increased from 43.5% (CI: 40.9%-46.1%) in Survey 1 (December 2020) to 81.1% (95% CI: 74.4%-87.8%) in Survey 2 (January 2022; $\chi2 (1) = 104.72$, $p < .001$; Fig 1).

## Vaccine hesitancy

A Pearson´s Chi-Squared test revealed a significant association (Fig 1) between time and vaccine hesitancy ($\chi2 (1) = 52.23$, $p < .001$), in which overall hesitancy increased from 32.5% (CI: 29.9%-35.1%) in Survey 1 (December 2020) to 58.0% (95% CI: 51.3%-64.7%) in Survey 2 (January 2022).

Table 3 shows the combined logistic regression model of factors contributing to COVID-19 vaccine hesitancy. Governmental mistrust was the strongest predictor of hesitancy in the model (OR: 2.90; 95% CI: 2.23–3.79; $p < .001$) (Fig 1). Further, participants who indicated agreement or uncertainty with at least one conspiracy belief (i.e., participants who ticked "yes" to indicate agreement or "I don't know" to indicate uncertainty) predicted greater vaccine hesitancy compared to participants who did not indicate agreement or uncertainty (OR: 1.36; 95% CI: 1.07–1.72; $p = .010$).

There were no significant predictors of vaccine hesitancy among participants who used the health service (OR: 1.15; 95% CI: 0.91–1.45; $p = .25$), the mass media (OR: 0.74; 95% CI: 0.47–1.16; $p = .189$), or government officials (OR: 0.81; 95% CI: 0.64–1.04; $p = .095$) for COVID-19 vaccine-related information compared to participants who reported not using these platforms. However, participants who reported using internet webpages (e.g., news websites, blogs, Google) as a source of vaccine-related information were significantly more likely to report vaccine hesitancy than those who did not use the internet (OR: 1.39; 95% CI: 1.06–1.82; $p = .016$).

Finally, there were several significant demographic and socio-demographic factors. Greater hesitancy was observed among Muslim participants compared to Christian participants (OR:

**Table 2. Breakdown of COVID-19 misinformation beliefs in Surveys 2 and 3.** *Note*: Percentages may not equal 100 due to incomplete questions.

| | Combined (N = 1641) | 2020 (N = 1429) | 2022 (N = 212) |
|---|---|---|---|
| | % (n) | | |
| **A biological weapon designed by the government of China** | | | |
| Yes | 9.6 (157) | 8.2 (117) | 18.9 (40) |
| No | 59.0 (969) | 59.6 (852) | 55.2 (117) |
| I don't know | 31.4 (515) | 32.2 (460) | 25.9 (55) |
| **A virus designed by the pharmaceutical companies to sell their drugs** | | | |
| Yes | 5.4 (89) | 2.9 (41) | 22.6 (48) |
| No | 58.0 (951) | 62.7 (896) | 25.9 (55) |
| I don't know | 36.6 (601) | 34.4 (492) | 51.4 (109) |
| **An exaggeration by news media to cause fear and panic** | | | |
| Yes | 8.9 (146) | 7.7 (110) | 17.0 (36) |
| No | 64.7 (1061) | 68.9 (984) | 36.3 (77) |
| I don't know | 26.4 (434) | 23.4 (335) | 46.7 (99) |
| **A plague caused by sins and disbelief in human beings** | | | |
| Yes | 15.9 (261) | 15.9 (227) | 16.0 (34) |
| No | 61.9 (1016) | 63.4 (907) | 51.4 (109) |
| I don't know | 22.5 (365) | 20.7 (296) | 32.5 (69) |
| **Designed to reduce or control the population** | | | |
| Yes | 9.2 (151) | 8.3 (118) | 15.6 (33) |
| No | 57.0 (935) | 57.7 (824) | 52.4 (111) |
| I don't know | 33.3 (555) | 34.1 (487) | 32.1 (68) |
| **A biological weapon designed by the US government** | | | |
| Yes | 5.6 (92) | 1.3 (18) | 34.9 (74) |
| No | 59.5 (977) | 63.1 (901) | 35.8 (76) |
| I don't know | 34.9 (572) | 35.7 (510) | 29.2 (62) |
| **A result of 5G technology being installed in the country** | | | |
| Yes | 2.4 (40) | 1.1 (16) | 11.3 (24) |
| No | 65.5 (1075) | 68.2 (974) | 47.6 (101) |
| I don't know | 32.1 (526) | 30.7 (439) | 41.0 (87) |
| **Transmitted by radio waves** | | | |
| Yes | 0.5 (8) | 0.1 (2) | 2.8 (6) |
| No | 74.8 (1227) | 78.1 (1116) | 52.4 (111) |
| I don't know | 24.7 (406) | 21.8 (311) | 44.8 (95) |

1.33; 95% CI: 1.05–1.69; *p* = .018), and significantly lower hesitancy among participants living in the Kara (OR: 0.58; 95% CI: 0.38–0.90; *p* = .015) and Savannah (OR: 0.45; 95% CI: 0.26–0.76; *p* = .003) regions of Togo compared to Lomé. There was also marginally higher hesitancy among females compared to male participants (OR: 1.23; 95% CI: 0.97–1.53; *p* = .080) (Fig 2).

## Discussion

This study describes evidence of changes in overall levels of vaccine hesitancy in Togo across two points in time during the COVID-19 pandemic response. Hesitancy increased between December 2020 and January 2022. Survey 2 was after COVID-19 vaccines became available in Togo, with the associated health promotion activity that came with the vaccine roll out. Belief or uncertainty around key conspiracy theories also increased between surveys 1 and 2. Among key groups more likely to express hesitancy included Muslims, females, individuals who

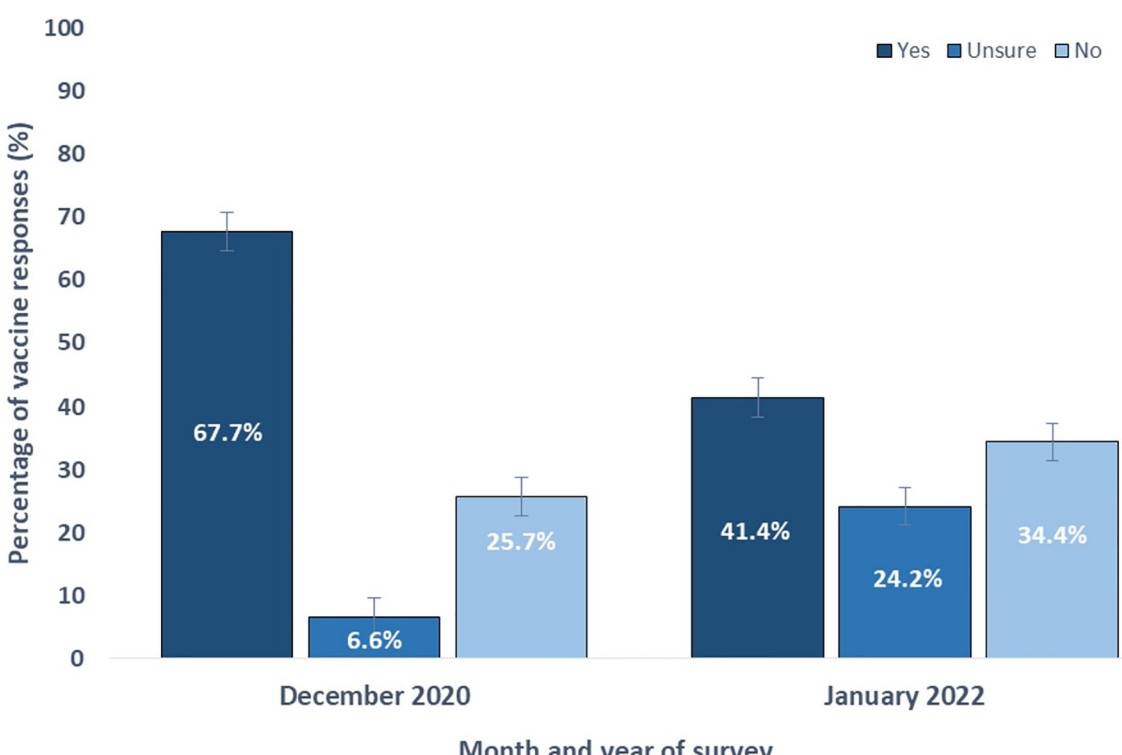

**Fig 1. Breakdown of yes, no, and unsure responses across two nationally representative surveys conducted in Togo in December 2020 and January 2022.**

**Table 3. Combined logistic regression model of factors contributing to COVID-19 vaccine hesitancy.** $N = 1642$. $R^2 = 0.110$.

|  | OR | p-value | 95% CI |
|---|---|---|---|
| Aged 40+ (ref. = 18–39) | 1.097 | .446 | 0.865–1.390 |
| Women (ref. = men) | 1.225 | .080 | 0.976–1.538 |
| Region (ref. = Lomé) |  |  |  |
| Central | 0.709 | .118 | 0.461–1.091 |
| Kara | 0.582 | .015 | 0.377–0.899 |
| Maritime | 1.130 | .416 | 0.842–1.516 |
| Plateaux | 0.955 | .788 | 0.681–1.339 |
| Savannah | 0.450 | .003 | 0.264–0.765 |
| Higher education (ref. = lower) | 0.866 | .285 | 0.665–1.128 |
| Islam faith or other (ref. = Christianity) | 1.333 | .018 | 1.051–1.689 |
| Governmental mistrust (ref. = low mistrust) | 2.904 | < .001 | 2.227–3.788 |
| Care responsibilities for adults or children (ref. = none) | 0.880 | .302 | 0.689–1.122 |
| Conspiracy beliefs or uncertainty (ref. = none) | 1.359 | .010 | 1.074–1.719 |
| Channels of COVID-19 information |  |  |  |
| Mass media (e.g., radio, newspapers, TV) | 0.738 | .189 | 0.468–1.162 |
| Togo Health Service or health workers | 1.145 | .254 | 0.907–1.446 |
| Government officials | 0.812 | .095 | 0.636–1.037 |
| Internet (e.g., Google, news websites, blogs) | 1.390 | .016 | 1.063–1.818 |

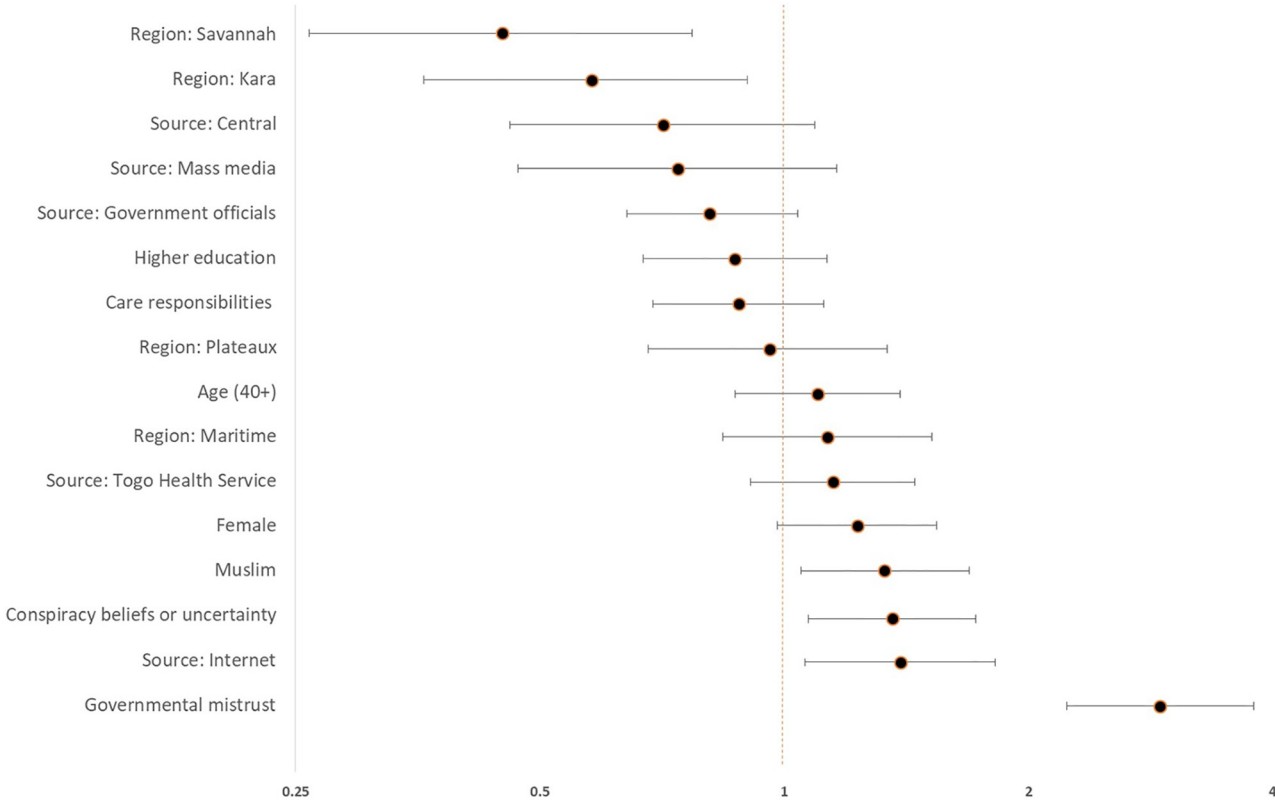

**Fig 2. Combined logistic regression model of factors contributing to COVID-19 vaccine hesitancy (N = 1138, R2 = 0.144).**

received COVID-19 information from internet sources, and individuals who expressed uncertainty about commonly-circulated COVID-19 misinformation beliefs. These findings were previously disseminated to Togo health stakeholders for their use in any pandemic-related health promotion or health system strategy.

Our results here show that presence of conspiracy theories are associated with public perceptions of vaccination. There is no 5G technology set up in Togo–despite this, around one-third of participants here indicated they were unsure about whether 5G was responsible for the spread of COVID-19. During the pandemic, the term 'infodemic' was used to describe the prevalence of both too much, and false, information which resulted in, for example, the WHO African Region setting up an Infodemic Response Alliance [16]. The scope of this Alliance is to monitor and proactively counter misinformation, conspiracy theories and rumours, thus supporting health service stakeholders with their engagement with the general public. The monkeypox public health emergency also saw an 'infodemic', with significant quantities of misinformation [17]. This can have a negative impact on any health systems response with individuals feeling stigmatised and reluctant to seek healthcare.

Trust in governance has emerged as an important factor around public health messaging and infection control during the COVID-19 pandemic [18]. Our results from Ghana showed that political allegiance has a role to play in confidence around vaccination [11]. Where the government is the source of any health promotion messaging, this may be more willingly received by populations who voted for them. Opposition voters may put less trust in that messaging. Our findings here add to the evidence base around the need for careful consideration

of who communicates the public health messaging, and how it can have the best possible impact. For example, where specific populations have lower trust in the government but high trust in the health service, the 'lead messengers' may best be healthcare workers. Equally, the use of high-profile individuals, who are popular and trusted, could help satisfactorily convey public health messaging. These individuals might include Chiefs, tribal Kings or Queens, or even celebrities. Clearly, there needs to be local strategies that are updated over time to reflect evolving socio-economic dynamics. Beyond the pandemic, vaccine confidence around routine immunisation will need to be monitored and new avenues explored around how to counter any misinformation.

Within one year of their implementation, the COVID-19 vaccines are estimated to have averted between 15-20m deaths globally [19]. For Togo, across 2020 and 2021, there were 248 confirmed deaths. However, modelling estimates excess death numbers of around 9030, a ratio difference of 36.4 between confirmed and excess mortality [1]. There undoubtedly will be a significant number of COVID-19 deaths that will never be recognised in official statistics. A lack of infrastructure to collect routine data, including birth and death registers, mean that many people in lower-income settings, including Togo, never make it into official statistics [20]. Given these limitations, it is very difficult to provide accurate real-time information during a public health emergency, such as the COVID-19 pandemic. Ahead of the next pandemic or high-profile outbreak, the implementation of improved routine data collection systems would allow for better-informed policy and planning.

One of the strengths of our study is the relatively large number of respondents, and that it covers two time points, rather than a one-off snapshot of public opinion. These time points also covered pre- and post- the introduction of the COVID-19 vaccines, so thus can give some indication around 'before and after' viewpoints. Further time-points would have allowed us to more closely monitor any evolving temporal trends. However, the need for internet or telephone access may have limited how representative the sampled population are to some extent. Certain groups will be under-represented, including those who reside in rural areas and people of lower socio-economic status–the so-called Last Mile populations [21]. There will likely be some response bias, given respondents were required to consent and take part via random selection from a phone call or have seen the internet survey and chosen to complete it. Phone coverage in Togo is thought to cover 90% of the population; [22] thus here, our methodology has the potential to access the majority of the country. Our shortfall in reach likely to be in the 'Last Mile' populations in hard-to-reach and under-served rural locations, and these are important groups who often lack a voice in healthcare and decision-making. Further studies should consider how best to integrate their feedback into policymaking. There may also be biases in terms of those who agreed to take part versus those who did not answer the calls or did not consent to participation. Sharing of mobile phones among household or communities further introduces potential biases. However, telephone surveys remove the cost of travel and allow access to those spread across a large geographical areas, such as Togo. Therefore, this combined with an inability to conduct face-to-face interviews in the 1st survey due to health concerns and travel restrictions, means that it is the most appropriate method for this research.

When considering the participants who were unvaccinated in Survey 2, we fell short of the required sample size. However, at the times the survey was completed, there were national guidelines in place around population movements, and thus remote approaches were more feasible, locally-acceptable, and more cost effective than in-person data collection around specific communities. We were also able to gain responses nationally via these methods, which may have been less feasible with data collectors on the ground.

## Conclusion

Hesitancy rates among unvaccinated individuals in Togo increased between time December 2020 and January 2022. Among key groups more likely to express hesitancy included those who had a strong mistrust of government and belief in at least one key conspiracy theory around COVID-19 vaccination. Health promotion campaigns should use locally and nationally trusted knowledge providers (e.g., the health service) and distribute the public health messaging via trusted individuals, such as religious leaders. Messaging should also target media platforms that are used by hesitant population groups. There needs to be an awareness of the range and strength of the circulating misinformation, with proactive health promotion approaches to counter the misinformation. For example, campaigns can focus on addressing concerns about vaccine safety and side effects. These approaches can improve uptake of any COVID-19 vaccines within Togo. Key future research should focus around knowledge gaps created by the pandemic, such as addressing the impact of misinformation upon routine immunisation uptake, repeated cross-sectional surveys to review temporal trends in vaccine confidence, and how best to proactively educate key stakeholders about emerging misinformation in their communities.

## Supporting information

**S1 Text. French translation of the submission manuscript.**
(DOCX)

**S2 Text. Survey questions answered by the participants.**
(DOCX)

**S1 Fig. Map of Togo, sourced from Nations Online.** https://www.nationsonline.org/oneworld/map/togo-administrative-map.htm.
(DOCX)

## Acknowledgments

We acknowledge and thank the survey participants and data collectors for their time and assistance with this research.

## Author Contributions

**Conceptualization:** Ken Brackstone, Jean-Paul Fantognon, Michael G. Head.

**Data curation:** Herve Akinocho.

**Formal analysis:** Ken Brackstone, Nia Eastment.

**Methodology:** Herve Akinocho, Ken Brackstone, Jean-Paul Fantognon, Michael G. Head.

**Project administration:** Herve Akinocho, Ken Brackstone, Michael G. Head.

**Supervision:** Herve Akinocho.

**Writing – original draft:** Ken Brackstone, Michael G. Head.

**Writing – review & editing:** Herve Akinocho, Ken Brackstone, Nia Eastment, Jean-Paul Fantognon, Michael G. Head.

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
