## [Decision Letter · Decision Letter 0]

16 Oct 2023

PGPH-D-23-01633

COVID-19 vaccine hesitancy and conspiracy beliefs in Togo: Findings from two cross-sectional surveys

Dear Dr. Head,

Thank you for submitting your manuscript to PLOS Global Public Health. After careful consideration, we feel that it has merit but does not fully meet PLOS Global Public Health’s publication criteria as it currently stands. Therefore, we invite you to submit a revised version of the manuscript that addresses the points raised during the review process.

We look forward to receiving your revised manuscript.

Kind regards,

Humayun Kabir

Academic Editor

Journal Requirements:

1. Please provide separate figure files in .tif or .eps format only and remove any figures embedded in your manuscript file. Please also ensure all files are under our size limit of 10MB.

Additional Editor Comments (if provided):

Reviewers' comments:

Reviewer's Responses to Questions

**Comments to the Author**

1. Does this manuscript meet PLOS Global Public Health’s publication criteria? Is the manuscript technically sound, and do the data support the conclusions? The manuscript must describe methodologically and ethically rigorous research with conclusions that are appropriately drawn based on the data presented.

Reviewer #1: Yes

Reviewer #2: Partly

Reviewer #3: Yes

Reviewer #4: Partly

Reviewer #5: Yes

Reviewer #6: Partly

Reviewer #7: Partly

Reviewer #8: Yes

2. Has the statistical analysis been performed appropriately and rigorously?

Reviewer #1: Yes

Reviewer #2: Yes

Reviewer #3: Yes

Reviewer #4: No

Reviewer #5: No

Reviewer #6: Yes

Reviewer #7: Yes

Reviewer #8: N/A

3. Have the authors made all data underlying the findings in their manuscript fully available (please refer to the Data Availability Statement at the start of the manuscript PDF file)?

Reviewer #1: Yes

Reviewer #2: No

Reviewer #3: Yes

Reviewer #4: Yes

Reviewer #5: No

Reviewer #6: Yes

Reviewer #7: Yes

Reviewer #8: Yes

4. Is the manuscript presented in an intelligible fashion and written in standard English?

Reviewer #1: Yes

Reviewer #2: Yes

Reviewer #3: Yes

Reviewer #4: Yes

Reviewer #5: No

Reviewer #6: Yes

Reviewer #7: Yes

Reviewer #8: Yes

5. Review Comments to the Author

Reviewer #1: This paper reads very well and is sound in approach and presentation. I enjoyed reading it and the authors should be commended for a great piece of work.

Minor Revisions:

Methodology: More clarity is required regarding how the participants for each questionnaire were recruited. The questionnaire appears to have been conducted via telephone or via a webpage but it is unclear how participants accessed or became aware of the questionnaire.

Line 288 - Re Limitations of the study. Reference to the sample size of survey 2 not meeting the the calculated power number should be made.

Reviewer #2: This study conducted an analysis of vaccine hesitancy among citizens of Togo, utilizing data obtained from two national surveys. The topic holds significance for the epidemiological study of COVID-19. However, there are certain areas that require clarification. This paper can be considered for publication in PLOS Global Public Health after a comprehensive revision.

My specific comments are:

1. The purpose and innovations of this paper should be highlighted in the abstract and Introduction. Moreover, please condense the abstract section to include only the most important results and findings.

2. Page 46-47: Please provide the most recent information on the ongoing COVID-19 pandemic.

3.Page 4, lines 81-84: Please add a reference for this sentence.

4. Page 4, lines 86-89: You mentioned that you’ve already published two studies on vaccine hesitancy in Ghana. Rather than simply changing the location, what are the paper’s innovations?

5. Methods: Design, participants, and procedure: The surveys were conducted between 1-16 December 2020 and 11-28 January 2022. However, researchers may be interested in a more recent survey. Please include the most recent information.

6. Methods: There were 1,430 participants in Survey 1 and 212 participants in Survey 2, respectively. Why are the participant numbers so disparate? Would this disparity impact the reliability of the results?

7. Figures: It is recommended that all the Figures be redrawn due to their lack of clarity.

8. Please explain more about how this paper’s findings contribute to the comprehension and prevention of the COVID-19 pandemic.

Reviewer #3: The manuscript is well written and focuses on an important global health problem of acceptability of vaccines by populations to control infectious diseases. While statistical analysis was done well i want to bring to your attention Table 3. It seems channels of communication was not compared to a baseline. Also, by doing a multivariable statistical analysis, it is important to do a model assessment that selects the best model to use to explain your results.

The discussion section did not delve into potential biases from using telephone surveys and how the analysis might have accounted for these biases to strengthen associations observed. it is not good practice to change a likert scale response to a categorical response because a response of not sure is a standalone response.

Reviewer #4: The methodology section needs some improvement.

Sample size: Author should justify the chosen sample size, such as effect size. Also, state how the shortfall in survey 2 was addressed.

Ethical approval: Authors stated ethical approval. However, there is a need to include how patient confidentiality and privacy were ensured during data collection and storage.

Instruments: Authors should expand on the validity and reliability of the measures used, particularly for assessing vaccine hesitancy, governmental mistrust, and conspiracy beliefs. Also, provide details of the psychometric properties of these measures and any prior validation studies conducted in similar contexts

Analysis: provide a rationale for covariates in bivariate logistic regression analyses, including potential confounders. Also, explain the process of handling missing data to ensure data quality.

The author should provide a brief description of conspiracy beliefs to ensure clarity for readers who may not be familiar with them.

Also, on the association of time and conspiracy belief, the author should provide exact values including confidence intervals can also provide important information about the precision of the estimates.

Discussion section

251: The result shows the presence of conspiracy theories can impact public perceptions of vaccination. Kindly clarify if the analysis established a causal relationship or if this is an inferred association based on the findings.

The author stated, "the importance of trust in governance and political allegiance in influencing vaccine confidence"(268) Expand on these implications and provide specific recommendations for public health officials on communicating with different population groups and effectively addressing misinformation.

The potential limitations such as underrepresentation (282) and response bias. Provide a detailed discussion of these limitations and suggestions for future research to overcome these challenges. Additionally, discuss any other potential study design limitations or data collection methods.

Reviewer #5: Regarding statistical analysis both bivariate and multivariate( COR and AOR) should be computed to identify factors associated your independent variable.

The inclusion and exclusion criteria should clearly stated

What do you think the importance of conducting dual cross sectional study?

All required files should be presented in form of supplementary material

Reviewer #6: interesting manuscript

however, there are some things that deserve commentary

in the intro the assertion that "There are likely to be many factors contributing to the lower mortality in West Africa, including a younger population, better preparedness and better management of outbreaks" is in my opinion unsustainable as the diagnostic data and statistics available in the region are poor to make such an observation.

it seems fundamental to me to detail the access to telephone services in the country, given that it is described as a poor country and with an important lack of resources, answering a telephone survey may have a marked bias of those who have the capacity to participate in the study.

it is not clear why the calculation of sample sizes was so different between the two surveys, requires clarification of the wording.

the results are expressed in the text and in the tables, which is redundant.

the results according to geographical regions are difficult to understand for the reader not familiar with Togo, a map could be attached or characteristics could be given to make this geographical classification relevant.

Reviewer #7: SECTION Abstract:

Page 2 Line 23 - The author states in the first line of the abstract "Estimates suggest that ...." - What are these estimates? Can this source be cited? For example, WHO estimates or whatever study suggested those estimates- please include the reference of these estimates.

SECTION Introduction:

Page 4 Line 79 - Capitalize COVID - maintain consistency.

Page 4 Line 81 - The author mentioned that studies suggest COVID-19 misinformation - Which studies suggest this? Cite these studies.

Page 4 Line 82 to 84 - The author mentioned a nationwide survey in Ghana that said COVID-19 was a biological weapon - cite this nationwide survey and include it in your references.

SECTION Methods:

Page 5 Line 108 - The author mentioned that the unvaccinated in survey 2 fell short of required number. - What are the consequences / limitations of this findings. Include it in your discussion section under limitations.

SECTION Discussion:

Page 17 Line 272 - punctuation correction - period and a comma.

SECTION Conclusion:

Page 18 Line 302 - The author mentions the timeline of the survey - December 2020 and January 2022 - please specify the reason for choosing this timeline. And the findings that justify this time frame? Did this timeline cause any bias in the study results? Mention the justification in the methods as well, and if there are any bias - mention that in the limitations or how you adjusted for it.

ADDITIONAL SECTION REQUIRED: Include a section or a paragraph on public health implications or include as a subgroup in conclusion. There is one line in conclusion that states key future research should focus on closing knowledge gaps. What kind of research are you expecting to be conducted based on the results of your survey? How do your results contribute to what kind of future research needs to be done? How would you fill the knowledge gaps - Elaborate on this? What channels would you employ to improve the health literacy rates?

Reviewer #8: All necessary comments regarding the manuscript guide and paper content have been attached for the authors' correction. Most of the comments are related to the use of acronyms , missing words within the manuscript, absence of major headings and lack of clarity on some components of the methodology.

I will suggest the authors include the dates they access grey literatures on their reference lists

6. PLOS authors have the option to publish the peer review history of their article (what does this mean?). If published, this will include your full peer review and any attached files.

**Do you want your identity to be public for this peer review?** For information about this choice, including consent withdrawal, please see our Privacy Policy.

Reviewer #1: No

Reviewer #2: No

Reviewer #3: No

Reviewer #4: No

Reviewer #5: No

Reviewer #6: No

Reviewer #7: **Yes: **Aiswarya Bulusu

Reviewer #8: No

---

## [Decision Letter · Decision Letter 1]

18 Dec 2023

PGPH-D-23-01633R1

COVID-19 vaccine hesitancy and conspiracy beliefs in Togo: Findings from two cross-sectional surveys

Dear Dr. Head,

Thank you for submitting your manuscript to PLOS Global Public Health. After careful consideration, we feel that it has merit but does not fully meet PLOS Global Public Health’s publication criteria as it currently stands. Therefore, we invite you to submit a revised version of the manuscript that addresses the points raised during the review process.

We look forward to receiving your revised manuscript.

Kind regards,

Humayun Kabir

Academic Editor

Journal Requirements:

b. If any authors received a salary from any of your funders, please state which authors and which funders.

Additional Editor Comments (if provided):

Reviewers' comments:

Reviewer's Responses to Questions

**Comments to the Author**

1. If the authors have adequately addressed your comments raised in a previous round of review and you feel that this manuscript is now acceptable for publication, you may indicate that here to bypass the “Comments to the Author” section, enter your conflict of interest statement in the “Confidential to Editor” section, and submit your "Accept" recommendation.

Reviewer #1: All comments have been addressed

Reviewer #2: All comments have been addressed

Reviewer #7: All comments have been addressed

2. Does this manuscript meet PLOS Global Public Health’s publication criteria? Is the manuscript technically sound, and do the data support the conclusions? The manuscript must describe methodologically and ethically rigorous research with conclusions that are appropriately drawn based on the data presented.

Reviewer #1: Yes

Reviewer #2: Yes

Reviewer #7: Yes

3. Has the statistical analysis been performed appropriately and rigorously?

Reviewer #1: I don't know

Reviewer #2: Yes

Reviewer #7: Yes

4. Have the authors made all data underlying the findings in their manuscript fully available (please refer to the Data Availability Statement at the start of the manuscript PDF file)?

Reviewer #1: Yes

Reviewer #2: Yes

Reviewer #7: Yes

5. Is the manuscript presented in an intelligible fashion and written in standard English?

Reviewer #1: Yes

Reviewer #2: Yes

Reviewer #7: Yes

6. Review Comments to the Author

Reviewer #1: A valuable piece of research and enjoyable read.

Reviewer #2: The authors have adequately addressed my comments. This paper can be accepted in its current form.

Reviewer #7: Thank you for making the edits.

One last concern - i am not sure if discussion and conclusion sections should include citations. These sections should be about your conclusion and findings about the strengths and limitations of the your own study and not cite other studies unless in area where you are trying to find similarities in results to previously existing studies which credit/ discredit the current study. Please make sure this is addressed.

7. PLOS authors have the option to publish the peer review history of their article (what does this mean?). If published, this will include your full peer review and any attached files.

**Do you want your identity to be public for this peer review?** For information about this choice, including consent withdrawal, please see our Privacy Policy.

Reviewer #1: No

Reviewer #2: No

Reviewer #7: No

---

## [Decision Letter · Decision Letter 2]

5 Feb 2024

COVID-19 vaccine hesitancy and conspiracy beliefs in Togo: Findings from two cross-sectional surveys

PGPH-D-23-01633R2

Dear Dr Head,

We are pleased to inform you that your manuscript 'COVID-19 vaccine hesitancy and conspiracy beliefs in Togo: Findings from two cross-sectional surveys' has been provisionally accepted for publication in PLOS Global Public Health.

Best regards,

Julio Croda, Ph.D, M.D.

Academic Editor

Reviewer Comments (if any, and for reference):

Reviewer's Responses to Questions

**Comments to the Author**

1. If the authors have adequately addressed your comments raised in a previous round of review and you feel that this manuscript is now acceptable for publication, you may indicate that here to bypass the “Comments to the Author” section, enter your conflict of interest statement in the “Confidential to Editor” section, and submit your "Accept" recommendation.

Reviewer #1: All comments have been addressed

Reviewer #7: All comments have been addressed

2. Does this manuscript meet PLOS Global Public Health’s publication criteria? Is the manuscript technically sound, and do the data support the conclusions? The manuscript must describe methodologically and ethically rigorous research with conclusions that are appropriately drawn based on the data presented.

Reviewer #1: Yes

Reviewer #7: Yes

3. Has the statistical analysis been performed appropriately and rigorously?

Reviewer #1: I don't know

Reviewer #7: Yes

4. Have the authors made all data underlying the findings in their manuscript fully available (please refer to the Data Availability Statement at the start of the manuscript PDF file)?

Reviewer #1: Yes

Reviewer #7: Yes

5. Is the manuscript presented in an intelligible fashion and written in standard English?

Reviewer #1: Yes

Reviewer #7: Yes

6. Review Comments to the Author

Reviewer #1: (No Response)

Reviewer #7: As long as you adhere to the Journal Guidelines, I am willing to accept this submission. Good Luck!

7. PLOS authors have the option to publish the peer review history of their article (what does this mean?). If published, this will include your full peer review and any attached files.

**Do you want your identity to be public for this peer review?** For information about this choice, including consent withdrawal, please see our Privacy Policy.

Reviewer #1: No

Reviewer #7: **Yes: **Aiswarya Bulusu
